# Challenges and Opportunities for Global Genomic Surveillance Strategies in the COVID-19 Era

**DOI:** 10.3390/v14112532

**Published:** 2022-11-16

**Authors:** Ted Ling-Hu, Estefany Rios-Guzman, Ramon Lorenzo-Redondo, Egon A. Ozer, Judd F. Hultquist

**Affiliations:** 1Division of Infectious Diseases, Department of Medicine, Feinberg School of Medicine, Northwestern University, Chicago, IL 60611, USA; 2Center for Pathogen Genomics and Microbial Evolution, Robert J. Havey, MD Institute for Global Health, Northwestern University, Chicago, IL 60611, USA

**Keywords:** SARS-CoV-2, COVID-19, genomic surveillance, molecular surveillance, public health intervention, epidemiology, global health

## Abstract

Global SARS-CoV-2 genomic surveillance efforts have provided critical data on the ongoing evolution of the virus to inform best practices in clinical care and public health throughout the pandemic. Impactful genomic surveillance strategies generally follow a multi-disciplinary pipeline involving clinical sample collection, viral genotyping, metadata linkage, data reporting, and public health responses. Unfortunately, current limitations in each of these steps have compromised the overall effectiveness of these strategies. Biases from convenience-based sampling methods can obfuscate the true distribution of circulating variants. The lack of standardization in genotyping strategies and bioinformatic expertise can create bottlenecks in data processing and complicate interpretation. Limitations and inconsistencies in clinical and demographic data collection and sharing can slow the compilation and limit the utility of comprehensive datasets. This likewise can complicate data reporting, restricting the availability of timely data. Finally, gaps and delays in the implementation of genomic surveillance data in the public health sphere can prevent officials from formulating effective mitigation strategies to prevent outbreaks. In this review, we outline current SARS-CoV-2 global genomic surveillance methods and assess roadblocks at each step of the pipeline to identify potential solutions. Evaluating the current obstacles that impede effective surveillance can improve both global coordination efforts and pandemic preparedness for future outbreaks.

## 1. Introduction

The emergence and rapid spread of SARS-CoV-2, the causative agent of COVID-19, has resulted in a global pandemic with over 620 million cases and 6.5 million deaths since late 2019 (as of 1 October 2022) [1]. Like all viruses, SARS-CoV-2 continually mutates during its spread, resulting in new viral variants that may have phenotypic differences in replication [2] and/or host–pathogen interactions [3]. Mutations that confer a selective advantage in a given environment may allow a specific variant to outcompete others and expand in the population. Variants that spread rapidly or pose a public health risk due to enhanced transmissibility, risk of severe disease, or immune evasion are designated as variants of concern (VOCs) by the World Health Organization (WHO) [4].

To track the emergence of new variants and assess their potential public health risks, researchers and public health experts largely rely on genomic surveillance. Genomic surveillance refers to the combinatorial efforts of epidemiological analysis, next-generation sequencing, and bioinformatics to identify relationships between viral genetic diversity and public health outcomes [5]. Genomic surveillance at the population level assists in the rapid identification of new emerging variants and the targeted deployment of mitigation strategies by public health entities to prevents their spread. Linking sequences with clinical and molecular data further enables the association of specific mutations with pathogenic outcomes or therapeutic efficacy, providing critical information for determining best practices in clinical care. Genomic surveillance has been used for decades to inform the public health response to a number of pandemic pathogens, including H1N1 influenza A virus, Ebola virus, Methicillin-resistant *Staphylococcus aureus,* Zika virus, and Human Immunodeficiency Virus [6,7,8,9,10]. However, recent advances in sequencing technologies and accessibility, coupled with the global scope and urgency of the COVID-19 pandemic, have resulted in more genomic surveillance for SARS-CoV-2 than any other pathogen to date.

Genomic surveillance strategies are generally comprised of clinical sample collection, viral genetic analysis, linkage to metadata (and possibly clinical) data, reporting, and communication with public health agencies for response and messaging (Figure 1). A robust and sustainable infrastructure for genomic surveillance systems to track SARS-CoV-2 variants around the globe has the potential to substantially reduce the burden of disease through the timely dissemination of data to optimize research priorities, therapeutics development, public health responses, and clinical care. Unfortunately, several limitations in the global genomic surveillance infrastructure have dampened this potential, resulting in long delays in data availability and inconsistent messaging. The purpose of this article is to review the current state of SARS-CoV-2 genomic surveillance, identify limitations in the pipeline, and highlight potential solutions that may help minimize the impact of future outbreaks.

## 2. Sample Collection

The first step of genomic surveillance is the collection of specimens harboring intact SARS-CoV-2 genetic material from COVID-19-positive individuals. Often, collected clinical specimens are in the form of residual, PCR-based diagnostic tests that likewise rely on the detection of viral RNA. Most SARS-CoV-2 specimens are collected from the upper respiratory tract as nasopharyngeal (NP) or oropharyngeal (OP) swabs stored in an RNA-compatible media, such as viral or universal transport media (VTM or UTM) [11]. While other anatomical sites can be sampled (i.e., the lower respiratory tract via bronchoalveolar lavage (BAL) washings or sputum), they are generally less accessible, available, and/or convenient [12,13]. Specimens are then cataloged, organized, and preserved for downstream genotyping. Long-term preservation requires the use of low-temperature freezers designed to maintain the long-term integrity of biological specimens. Overall, while this process of sample collection seems relatively straightforward, the practice of convenience-based sampling, increased prevalence of antigen testing, and unequal resource distribution can bias collection.

First, most genomic surveillance studies rely on convenience-based sampling methods to collect their specimens. Convenience-based sampling is a type of non-probability sampling wherein specimens are collected from individuals that are selected for convenience and not truly at random [14]. For example, if a genomic surveillance study relies on the collection of residual diagnostic specimens from a local health center, the specimens are likely to reflect the populations that the health center serves, which may or may not reflect the demographics of the population of the broader region. This can further result in oversampling of populations in regions with better surveillance infrastructure and resources [15,16]. While systematic investment in surveillance infrastructure and an evidence-based sampling strategy would be needed to resolve these issues long-term, sampling biases can be detected and addressed in studies that rely on convenience sampling. One common method to detect sample bias is to test for statistical differences in the descriptive statistics of the sampled and represented populations such as the mean, variance, and distribution of demographic variables [17]. If pulling a subset from a larger collection of specimens, researchers can address sampling bias using stratified random sampling to ensure that the observed cohort matches the geographic and demographic distribution of the true population [18,19]. Alternately, another method to minimize the effects of sampling bias is to use mathematical modeling to correct these effects during study analysis [18,20].

Second, the rise of at-home, antigen-based tests and the prevalence of testing centers that lack the resources for specimen biobanking or sequencing pose major challenges to unbiased specimen collection. Rapid, antigen-based tests do not require professional administration or interpretation and have been widely adopted in some countries and populations for self-diagnosis [21]. These individuals are often not encountered in the health system and so are not likely to be sampled in convenience-based methods [22]. Even some testing centers that do use a PCR-based test for SARS-CoV-2 RNA, including many commercial testing centers and community clinics, often lack the incentive or resources to save or share specimens for surveillance [23]. These discrepancies in testing facilities results in many specimens originating from larger medical centers that are incentivized to perform specimen collection and biobanking either for research purposes or due to pre-existing relationships with outside partners, such as public health departments. This can result in an underestimation of the overall incidence of infection and a significant bias towards at-risk individuals and those presenting with more severe disease. With increased vaccination rates and immunity gained from the previous infection, infected individuals are less likely to progress to severe COVID-19 or present for routine upper respiratory tract testing [24,25,26]. Asymptomatic individuals are especially unlikely to be sampled except in coincidental cases such as before travel, before a medical procedure, or following testing after a known exposure. To overcome these limitations, alternative methods of non-invasive viral detection, such as wastewater surveillance, have been used to assess the overall incidence and viral variant distribution. Wastewater surveillance involves the detection and sequencing of viruses in sewage systems following viral shedding in fecal matter [27]. Recent efforts have implemented wastewater surveillance to quantify viral loads, estimate the true incidence of infection in a population, and detect emerging or under-sampled variants [28]. The Netherlands was the first to implement this system to monitor under-reported cases as well as serve as an early warning signal for the emergence of new variants in real-time [29]. Though useful in estimating the prevalence, wastewater surveillance loses granularity in its inability to be linked with other data types and thus can only serve as a tool to detect broad trends.

Finally, collected specimens must be cataloged in biospecimen repositories, commonly referred to as biobanks, which present their own challenges. Specimen biobanks provide tremendous value in epidemiological surveillance by providing samples for the development of therapeutics [30] and diagnostics [30,31]. Although biobanking obstacles are not novel, the unprecedented volume of samples due to the high community incidence of COVID-19 will require a sustainable way to efficiently store informative samples without overburdening research facilities [16]. This is especially challenging in low- and middle-income countries (LMICs) that often lack the resources, infrastructure, and trained personnel for broad-scale biobanking [32,33]. Addressing concerns of equitable sample collection is crucial to prevent under-representation of the true SARS-CoV-2 genetic diversity landscape (see also Section 6.4 on Equity in Public Health Responses). Furthermore, problems in standardization and quality assurance may cause downstream problems in specimen quality. Guidelines for the storage and processing of samples are typically specific to each institution with no universal recommendations on storage containers or labeling schemas [34,35]. The European Virus Archive (EVA), a non-profit dedicated to biobanking viral products, has developed a grading system that defines the virus product quality in a specimen, but it is unclear how many other biobanks have adopted the same system [36,37]. Institutional biobanks tend to be siloed within a single center due to administrative burdens, such that linking and establishing networks of biobanks across institutions becomes even more difficult and reduces the potential of large-scale research efforts. Biobanking issues can be resolved through standardization and well-defined guidelines that include clarity on the processing, governance, and sharing of data that will ultimately expedite multi-center collaborations. Furthermore, identifying biobanks collecting similar specimens could establish networks of biobanks that could easily improve the effective sample size of the biobank. Efforts to coordinate institutional sampling and biobanking have been implemented by governmental entities and public health departments to varying degrees of success [38]. Together, these measures would ensure the sustainability and longevity of these biobanks and ensure that they can be repurposed for future studies.

## 3. Genotyping

The second step in genomic surveillance is to determine the virus’s genomic sequence or genotype. Depending on the set of mutations that a virus carries, it can be assigned to a specific clade or lineage using phylogenetic analyses. While a couple of distinct nomenclatures exist for the naming of SARS-CoV-2 lineages, the Phylogenetic Assignment of Named Global Outbreak Lineage (or PANGOLIN) nomenclature is one of the most widely used. This dynamic nomenclature was proposed in 2020 to assign SARS-CoV-2 lineages as they arose [39]. PANGOLIN lineages are particularly useful for tracking prospective transmission events and outbreaks, ultimately complementing other nomenclature tools that focus on clade designation [40]. Mutations linked with specific lineages, VOCs, or phenotypes (i.e., drug resistance or increased transmission) can inform risk assessments and policy.

SARS-CoV-2 genotyping can be performed using a variety of techniques including specialized reverse transcription-quantitative PCR (RT-qPCR) assays, Sanger sequencing of targeted amplicons, or whole-genome sequencing (WGS). Of these, WGS is the most informative as it can be used to detect novel mutations and define new, emerging variants, whereas the other approaches, such as RT-qPCR, are generally limited for use in the detection of or differentiation between known variants. Throughout the COVID-19 pandemic, WGS of SARS-CoV-2 isolates has provided an in-depth understanding of the genetic characteristics and evolution of the virus [41,42,43]. Paired with clinical metadata, sequencing information can be utilized to study the association between specific mutations or variants and clinical outcomes [44], diagnostic failure [45], and therapeutic/vaccine efficacy [46]. Viral genetic data combined with more traditional epidemiological methods have been furthermore used to track routes of transmission and identify risk factors for infection [47]. Despite the power of these tools, several challenges have limited their broad adoption, including variability in sequencing protocols, lack of bioinformatic expertise, and high cost.

In response to the far-reaching pandemic, a large number of next-generation sequencing (NGS) technologies and approaches have been adapted for WGS of SARS-CoV-2. Each technology has its strengths and weaknesses and relies on slightly different sequencing protocols. This has led to technology-specific differences in read mappability, genome coverage, and sensitivity, as well as precision of calling single nucleotide variants (SNVs), all of which may influence the confidence of genotype determination [48]. This has been particularly challenging for samples with low viral loads, which often cannot be sequenced at high coverage or with high confidence [49]. While new and improved protocols for low-copy-number sequencing are under continual development, this has resulted in knowledge gaps in how the virus is evolving late in disease course, in people with high levels of immunity (due to prior infection or vaccination), or in compartments with only low-level replication (i.e., in the brain or gut). An improved understanding of the benefits and drawbacks of different sequencing protocols can help researchers select the protocol that produces the most accurate results given their circumstances. For example, fresh samples often yield better coverage than frozen samples [48], but this can be resolved by deeper sequencing or reduced sample multiplexing when working from a cryopreserved biobank. Another alternative would be to use metagenomic shotgun or target-enrichment sequencing protocols, which may improve yield and reduce error, though these approaches are lower throughput and more expensive. Transparent reporting of sequencing methods, adoption of standardized protocols (i.e., the ARTIC protocol has been highly used for SARS-CoV-2 sequencing [50]), and requirements for robust quality control assessments would all improve the reliability of viral WGS.

Bottlenecks in the bioinformatic analysis of SARS-CoV-2 sequence data present yet another challenge. After sequencing, researchers use bioinformatics tools to identify mutation frequencies, assign lineages, and infer population dynamics [51]. However, the lack of experienced personnel and reliable computational infrastructure has resulted in significant delays to data reporting and limited application of the data in more complex analyses [41]. The lack of standardization in bioinformatics pipelines, such as the assembly of raw reads, also poses an issue. There is currently no “gold standard” for genome assembly, which results in errors that get incorporated in downstream analysis. There is an urgent need for easily implementable, interpretable, and reproducible bioinformatics pipelines that can be utilized with minimal training. Further studies may be warranted in assessing current bioinformatics pipelines, such as genome assemblers, to create a standardized pipeline that results in rapid analyses post-sequencing and generate actionable baseline results [51,52,53,54,55]. Point-and-click software is a class of software that is user-friendly and requires minimal domain knowledge for use. Nextstrain is one such example, where the software can take raw sequence data and offer flexible analyses that output interpretable results for researchers to understand the genome of their specific SARS-CoV-2 samples [56]. The disadvantage of point-and-click software lies in the restricted functions encoded in the software. Even with functional limitations, point-and-click software can serve as a solution in resource-limited settings. Another solution could be the use of installation-free cloud computing, where sequencing data can be uploaded onto a secure server, and results such as mutational profiles of raw sequences can be produced by a third party [57]. Regardless, the lack of reliable internet service and computing power in certain regions of the globe, particularly in LMICs, are limiting factors.

Other, more accessible approaches for SARS-CoV-2 genotyping have been developed that require less specialized equipment and expertise. Given the prevalence of RT-qPCR machines in most research settings, many institutions have used RT-qPCR protocols to identify specific SARS-CoV-2 variants. The RT-qPCR workflow is familiar and cost-effective as compared to most sequencing protocols and has been effectively employed to rapidly screen thousands of specimens for the emergence of specific variants. For example, when the Alpha VOC became predominant, a deletion in the Spike gene open reading frame (ORF) resulted in the loss of amplification for that target in diagnostic tests, a phenomenon called the S-Gene Target Failure (SGTF) [58]. SGTF is defined as a non-detection of the S-gene target in samples that test positive for both the N-gene and ORF1ab gene targets. Certain Omicron variants have also presented with SGTF due to internal Spike deletions, which have been used by some as a proxy for assigning the lineages without direct sequencing of the sample [59]. While high-throughput and less costly, the main limitation of RT-qPCR is that it can only detect known variants. Primer sets must be specifically designed to known portions of the genome that differ between previously described variants. As an alternate approach, less resource-intensive WGS protocols have been developed that may be suitable for application in resource-limited settings. For example, Oxford Nanopore Technology is a newer sequencing platform that can be manufactured at a cheaper rate compared with other high-throughput sequencing platforms. While this technology is less high-throughput, has a higher error rate compared to Illumina sequencing platforms, and produces longer but fewer reads, it is sufficient for SARS-CoV-2 WGS [60].

## 4. Clinical and Metadata Linkage

The third step in genomic surveillance is to link the viral genotype data with available demographic, epidemiological, and clinical metadata. While sequence data alone is useful for the analysis of viral evolution and for understanding macro-trends in viral population structures, its value to public health and clinical care comes from linkage to these additional datasets [16,61,62]. Comprehensive genomic surveillance data, including genomic, demographic, epidemiological, and clinical data are required to identify associations between specific viral lineages and patient risk factors, disease severity, immune escape, transmissibility, and therapeutic efficacy [4]. The establishment of comprehensive surveillance networks that utilize multiple data types to accurately assess risk will be a key tool to fight the continual spread of SARS-CoV-2 as well as future pathogens [63,64]. In the meantime, however, challenges with data compartmentalization, standardization, and administrative burdens will need to be addressed to best leverage ongoing genomic surveillance efforts.

The primary challenge in data linkage is often compartmentalization. Many of the data types discussed above are generated in different locations: public health departments generate epidemiologic data, research centers generate genomic data, and hospitals generate clinical and vaccination data. In a majority of cases, however, there is often no strategic coordination between these entities to link different data types together to form a comprehensive dataset [59,65]. Though monomeric or partial data types can and do provide valuable insight [66], data linkage is necessary for maximal utility and benefit. Even when data types are linked, oftentimes the data that are shared are limited, have a high percentage of missing values, and are only provided voluntarily. For example, while GISAID has become a prime example for the international sharing of SARS-CoV-2 sequence data, the metadata associated with each sequence are minimal, limiting interpretability [67]. The amount of available metadata varies substantially by country (Figure 2). The United States and the United Kingdom have generated and uploaded the largest number of sequences to GISAID, but only ~50% and <5%, respectively, have associated sex and age metadata. On the contrary, Slovakia and Slovenia have high levels of associated age and sex metadata but have provided limited sequence data relative to the rest of the world. One way to combat compartmentalization is to establish strategic interdisciplinary collaborations by identifying data management commonalities and creating flexible data use and sharing agreements that allow for the timely sharing of data while protecting patient privacy [68]. Another solution would be for widely adopted systems, such as GISAID or Genbank, to require minimally standardized, de-identified, and structured metadata requirements. Metadata standards such as the Minimum Information about any (x) Sequence (MIxS) as developed by the Genomic Standards Consortium (GSC) provide guidelines for what metadata should be shared with each sequence to enable an in-depth analysis of these sequences [69,70].

Another major issue is the lack of standardization in data-sharing practices. Interoperability in healthcare research has been a longstanding issue, but the COVID-19 pandemic brought this problem to the forefront [71,72]. Before the pandemic, the development of the FAIR (Findable, Accessible, Interoperable, and Reusable) data principles as well as the establishment of the Research Data Alliance (RDA) began the movement toward integrated data management practices [73,74]. However, the rapid onset of the pandemic exposed the lack of readiness and data-sharing infrastructure in our current health institutions. The variation of data management pipelines, inconsistencies in data fields, and lack of adherence to standardized record-keeping practices resulted in a lack of interoperability between and even within institutions that greatly complicated the compilation of comprehensive metadata [69]. Although there have been previous guidelines for the standardization of health information (i.e., HL7, FHIR, LOINC, SNOMED, ICD-9/ICD-10, etc.) [75], these guidelines are often ambiguous and are not widely implemented by hospital systems across different countries [76]. Even within a standardized data system, heterogeneous case definitions and differences in healthcare professional training could alter how a patient is an input into electronic health records [77,78]. Wider adoption of existing frameworks for interoperability could reduce issues of data linkage. For example, SNOMED (Systemized Nomenclature of Medicine) is a mature framework for clinical terminology that seeks to collate multiple, related medical concepts into one broader, interpretable medical term [79]. It is often used to overcome interoperability issues due to its ability to provide standardized terminologies [80]. New SNOMED terms could be adapted to COVID-specific symptoms that would allow for translatable knowledge between institutions [71]. It also provides a standardized dataset that could be leveraged for machine-learning-based algorithms or other analytical approaches to extract clinical insights.

Beyond compartmentalization and lack of interoperability, administrative and political roadblocks have hindered data sharing and/or record linkage. To protect patient privacy, different countries, regions, and institutions have different data management policies that can vary widely and can often present insurmountable barriers to data sharing. As one example, the European Union has adopted policies such as the General Data Protection Regulation (GDPR) that place firm restrictions on the international sharing of health data [81]. Policies that are too burdensome can hinder public health innovation and geographically restrict discoveries that must then be independently assessed on a global scale [82]. Many policies likewise require patient consent for even minimal data sharing. This is often not possible when working with retrospective biobanks while prospective sampling and consenting are resource-intensive. Furthermore, recent studies have found that an increasing number of patients have concerns over sharing their personal health information (PHI) [83], which ultimately results in the withholding of information from the provider and an unwillingness to consent to any data-sharing policies. Data sharing from LMICs is complicated further by a lack of trust, inequitable distribution of resources, and diminished returns for contributions [84,85]. In the long term, these issues could be addressed through the establishment of pre-existing protocols for patient data collection, consenting, and sharing as well as the development of international data-sharing consortiums that can easily share data across international borders. Additional guidance on global data sharing towards normalized standards for PHI handling and the protection of patient privacy would further minimize administrative burden. In the meantime, additional analysis methods are currently being developed to enable statistical modeling in the absence of direct data sharing. For example, federated learning is a new infrastructure to train machine-learning algorithms on data without the actual exchange of data itself [86]. The algorithm processes data locally and only model characteristics are transferred across different sites, hence, bypassing the data-sharing governance and privacy rules. In the context of genomic surveillance, models trained to predict severity could be transferred across institutions to be continually refined by the additive power of increased sample size. This model could prove to be very powerful and robust if it were to leverage existing data from multiple institutions across the globe.

## 5. Reporting

The fourth step in genomic surveillance is data dissemination and reporting to public health or international governing entities. The reporting hierarchy begins with research institutions and genomic surveillance centers that relay molecular findings to local public health organizations, disseminate sequence data through public repositories, and publish research studies in open-access journals. Ultimately, these data are collated and reviewed by national and transnational entities, such as the Centers for Disease Control (CDC) in the United States or the WHO, where the body of evidence is assessed by public health experts and translated into messaging and policy recommendations. Identifying components from successful pre-COVID-19 infectious disease reporting (i.e., timeliness, automation, completeness in data, etc.) has helped to define metrics to evaluate and strengthen our current surveillance systems. For example, a mixed-methods approach was used to evaluate the national reporting of epidemiologic data handled by the Indonesian Early Warning Alert and System from 2017 to 2019. This study identified completeness and timeliness as critical components to successful reporting, but this was contingent on robust infrastructure and technical expertise [87]. Many of these challenges persist including variability in the timeliness of reporting genomic data, overly centralized data flows that exclude self-reporting mechanisms, and ineffective communication strategies.

The timeliness of data reporting to relevant public health institutions during the pandemic has varied considerably across the globe, highlighting a key barrier to the real-time surveillance of pandemic threats. When assessing surveillance reporting using the CDC’s “Updated Guidelines for Evaluation of Public Health Surveillance” system, collection to submission time (CST) in Nigeria ranged from 24 h to two weeks in 2020–2021 [88]. However, other regions of the globe such as Hong Kong and Qatar reported much more variable CSTs ranging from 24 h to over 1 year. The reasons for delayed reporting within national surveillance systems can be attributed to infrastructural limitations, especially in LMICs, and the lack of a coordinated data reporting plan that fails to escalate local-level surveillance data. In contrast, many higher-income countries, such as the United Kingdom, have had consistently lower CSTs (as fast as three days for 76.6% of all cases) [89], which may be attributed to strong investments in genomic surveillance and a centralized public health system [90]. In general, we see a trend where higher-income countries tend to have shorter CSTs than LMICs (Figure 3). As a consequence of variable reporting, the delayed identification of emerging variants within populations with high CST leaves global efforts susceptible to poor preparation for new waves of cases from emerging variants. While increased funding, structured incentives, and defined reporting hierarchies may help to minimize CST, continued evaluation of surveillance systems using quantifiable and comparable metrics will be required to identify other key variables to improve the timeliness of genomic surveillance data reporting [91].

Most reporting workflows are centralized through clinical care centers and testing sites, but this structure relies on patient encounters that may not be representative of the broader population. Especially with the increased availability of at-home testing and decreased risk of severe disease due to COVID-19, the reliance on traditional reporting routes may underestimate the overall incidence and exclude critical segments of the population. Indeed, both traditional surveys and immunological screening suggest that the incidence of COVID-19 was higher than reported throughout the pandemic. One strategy to mitigate this would be to adopt strategies that enable self-reporting of infection and outcomes. Indeed, the use of machine learning to harvest informal self-reporting data has been explored as a method to improve accurate case estimations. A US retrospective study surveying self-identified COVID-19 infections on Twitter suggested higher-than-reported case counts, highlighting the need to account for non-canonical means for measuring incidence [92]. The use of self-reporting to expand contact tracing using mobile devices has also proved to be effective. For example, the Taiwan CDC’s Central Epidemic Command Center launched a self-report contact tracing system to relay information to epidemic prevention coordinators to limit community spread [93]. In addition, self-reporting could be leveraged to complement or reinforce current reporting systems by tracking patients’ post-treatment or re-infection status. Although self-reporting would assist in the accurate estimation of case counts and targeted follow-up, many genomic and clinical gaps in understanding would remain dependent on health system encounters.

Beyond variable timeliness and rigid reporting workflows, effective communication of surveillance data to a non-expert audience remains a critical challenge to a successful reporting strategy. The messaging platforms used to relay surveillance information can broadly be categorized into: (1) platforms curated by and targeted towards specialists for SARS-CoV-2 information (i.e., GISAID, Nextstrain, or Pubmed), (2) data tracking and messaging platforms curated by specialists and targeted towards general audiences (i.e., the New York Times Data Tracker, CDC website/press releases, the John Hopkins Coronavirus Resource Center, the Yale SARS-CoV-2 Genomic Surveillance Initiative, etc. [94]), and (3) social media platforms that facilitate conversations about SARS-CoV-2 information among the public. While social media platforms such as Twitter have proven to be a feasible outlet for public health messaging given appropriate monitoring and sufficient public engagement [95], they have also enabled the spread of misinformation that has greatly complicated public health responses in some countries. The expansion of mobile phone network messaging has also proven to be an effective tool for reporting and public health messaging in response to COVID-19 and other infectious disease outbreaks [96], especially in LMICs [97,98]. Lastly, communication among molecular surveillance experts in different sectors plays a crucial role in refining and relaying accurate public health information. One notable successful collaboration between those performing molecular surveillance and public health figures is in the COVID-19 Genomics UK (COG-UK) consortium launched in March 2020 [63]. This consortium collaborated with the National Health Service (NHS) and the Scientific Advisory Group for Emergencies (SAGE) to generate weekly datasets that focused on presenting digestible datasets to multiple stakeholders. Overall, successful communication—even among trained molecular surveillance specialists—is pivotal to streamlining reporting efforts.

## 6. Public Health Response

The fifth and final step in genomic surveillance is to inform an effective public health response. Technological and bioinformatics advances in pathogen genomics present a unique opportunity to utilize molecular epidemiology for the enhancement of public health responses. In the COVID-19 pandemic, effective genomic surveillance pipelines have proven to be essential in promoting evidence-driven public health messaging and interventions. Ideally, an all-encompassing, surveillance-informed intervention strategy would reflect benefits across the essential public health services framework [99]. This framework is composed of actionable steps to achieve effective strategies that indiscriminately promote health across all populations and emphasizes four focus areas: (1) assessment, (2) policy development, (3) assurance, and (4) equity (Figure 4). Previous scholars have documented how inequities in genomic surveillance during the COVID-19 pandemic can be attributed to the lack of foundation and/or support in one or more of these essential service areas [99]. Specifically, these studies have pointed to the negative influence of neglected infrastructure, siloed institutions with disjointed public health messaging, and improper intervention strategies to mitigate the spread of infection on genomic surveillance efforts. Furthermore, the lack of surveillance in resource-limited countries and the lack of infrastructure to support this work in under-sampled populations highlight areas that could benefit from international collaboration and capacity-building efforts. To simplify this discussion, we will explore current limitations using the public health services framework and evaluate current ongoing efforts that incorporate molecular surveillance with public health assessment and policy development to increase equity in public health responses to disease outbreaks.

### 6.1. Assessment

Essential public health services start with assessment, including assessing risk, identifying high-risk populations, and understanding public health behaviors that increase transmission. Assessing the global disease burden of COVID-19 and identifying risk factors that increase the incidence or severity of disease rely heavily on molecular diagnostics and genomic surveillance. Without proper assessment, all downstream actions are limited and may not target appropriate audiences. Early in the pandemic, clinical, demographic, and epidemiological datasets were leveraged to identify routes of transmission and high-risk populations for infection and severe disease outcomes (i.e., older age, higher body mass index, etc.). Transmission models and identification of high-risk populations were then incorporated into public health assessments to inform best practices. For example, surveillance and modeling studies in national [100] and multi-national [101] cohorts were crucial to inform public health policy on how to limit transmission in long-term care facilities (LTCFs) and healthcare workers (HCWs). The WHO’s Mass Gathering COVID-19 Risk Assessment tool leverages real-time epidemiology data and genomic surveillance information to enable risk evaluation, recommend mitigation measures, and suggest communication strategies [102].

Periodic reassessment of risk as new viral variants emerged was critical to inform the implementation of mitigation measures and public health policies designed to limit disease burden. For example, the increased transmissibility of the Delta variant resulted in revised policy recommendations for indoor masking [103,104,105,106] while enhanced immune escape by the Omicron variant resulted in updated vaccine recommendations.

Low-socioeconomic-status communities have limited access to healthcare [107], which subsequently limits access to testing. As a result, these communities were under-sampled and are under-represented in surveillance efforts [108]. Under-sampling of these populations could negatively bias results to exclude the effects of social determinants that are correlated with disease severity and to skew public health responses due to inaccurate representations of the population [109,110]. Moving forward, increasing gaps in sequencing data availability for multi-center modeling, especially in low-income communities, are likely to impede further public health assessment [111].

### 6.2. Policy Development

Surveillance systems aim to provide accurate and high-quality data in real-time to inform multiple stakeholders with the hope of influencing public health policy development. However, the timeliness in which surveillance information is reported to appropriate agencies and the public can have a large influence on the effective implementation of policy and public opinion [112]. Likewise, metadata completeness and the interconnectedness of surveillance systems play critical roles in informing policy. These components have previously been reported to be obstacles in prior surveillance systems globally, and have been re-identified during the COVID-19 pandemic as an area requiring improvement [113]. The mode of health messaging dissemination is as important as the quality of the surveillance data being reported [114]. Public health briefings, social media platforms, and traditional media such as news stations and radio are common ways in which public health recommendations are disseminated to the public and strengthen policy development [115].

Several key instances of COVID-19 surveillance influencing policy development showcase its potential to limit incidence. In Germany, routine national surveillance overlapping with periods of non-pharmaceutical interventions (NPIs), such as restrictions on mass-gathering events and targeted public lockdowns, demonstrated real-time disruption in transmission dynamics [116]. Critically, assessing the impact of NPIs on infectious disease transmission was dependent on ongoing surveillance. Similarly, changes in public policy can have impacts on the effectiveness of genomic surveillance strategies and reported incidences. For example, in July 2020, contact tracing policies in South Sudan were relaxed to test only symptomatic cases, resulting in substantial under-reporting of incidence [117]. Different methods of public health governance, whether they be centralized or decentralized (i.e., unilateral vs. shared decision-making), also impact how public health policy is implemented. The US has distributed public health governance through approximately 59 health departments with mixed centralization models [118], whereas other nations such as Australia have a centralized and unilateral decision-making process.

The strong association between public perception and its influence on public policy development and adherence has been documented in multiple accounts [119,120,121]. By far, the largest contributor to the public perception of COVID-19 relies on the pillar of informing, educating, and empowering individuals to engage in safe practices. On a global scale, this requires targeting a broad range of stakeholders in addition to the public. Improper surveillance can skew reported case incidences or result in spurious associations that can result in a cascade of misinterpretation and inaccurate data dissemination. During the COVID-19 pandemic, the spread of misinformation has been amplified through digitally enabled platforms in what is known as an “infodemic” [122]. Future strategies to avoid molecular surveillance misinformation should involve partnerships with social media outlets to aggregate data on misinformation spread [123].

### 6.3. Assurance

Public health assurance encompasses the enforcement of policies, linking policies to care, promoting a well-informed workforce, and evaluating current strategies [124]. Assurance can be further described as the reinforcement of developed policies and evaluation of whether implemented changes and surveillance systems met their objectives. Therefore, public health assurance requires a constant reassessment of molecular surveillance objectives, methods, and technology to optimize data-to-public health benefits. Overall, assurance, as it relates to SARS-CoV-2 genomic surveillance, is ongoing as contextualized by the roadblocks and limitations present at each stage discussed above. It should be noted that the routine monitoring of surveillance performance has historically strengthened other systems, such as the National Human Immunodeficiency Virus (HIV) Surveillance System and the National Notifiable Disease Surveillance System (NNDSS) in the United States [125].

### 6.4. Equity

Global genomic surveillance efforts are not immune to exclusionary practices that prevent equitable collaboration and representation. Prioritization of equity in both global surveillance and the public health response is required to achieve inclusive and sustainable impacts on the broader community. Equity in global genomic surveillance involves identifying vulnerable and high-risk populations that are either (1) under-sampled in surveillance systems or (2) more susceptible to worsened clinical outcomes. As previously mentioned, variant under-sampling is a current obstacle that prevents us from accurately understanding the viral landscape in different populations. Similarly, under-sampling of vulnerable and high-risk populations undermines public health intervention effectiveness, as under-reporting can result in a delay in identifying reservoirs of infection or emerging variants. The Economic Community of Central African States (ECCAS), composed of 11 central-African nations, found that only 0.9% of all reported cases were actively sequenced and reported from the region [126]. The current sampling of these populations is not representative of these central-African nations, which resulted in the delayed reporting of the emergence of the Eta variant in the region in late 2020 [127]. Even within the United States and the United Kingdom, the countries that have contributed the most sequences to the GISAID repository, there are geographic locations that are under-sampled in proportion to their population (Figure 5). This is reflective of intranational inequities and policies that require monitoring and active intervention to bolster an equitable public health response.

Achieving equity in the context of international research during the SARS-CoV-2 pandemic requires proper evaluation of key obstacles that can hinder surveillance. Funding, authorship, trust, recognition, and infrastructure support are recognized as critical components of successful international collaboration [128]. Respect for cultural beliefs and practices in international collaboration must also hold equal importance as the technical aspects needed for research and public health interventions [129]. As opposed to relying on outsourcing technical expertise and resources to high-income countries, collaborations such as the COVID-19 Clinical Research Coalition (CRC) strive to equip LMICs to perform genomic surveillance in their countries [130]. This instance of greater capacity building requires financial assistance from external sources including international agencies, academic institutions, and industrial partners. Collaborative efforts that do not equip LMICs with infrastructural support or building capacity for local researchers to drive surveillance and evidence-driven public health fall short of sustainable and mutually beneficial collaboration. Overall, equity in the context of both the population at risk for COVID-19 as well as stakeholders involved in surveillance and population-level interventions must be considered.

## 7. Future Directions and Conclusions

The scope and far-reaching impacts of the COVID-19 pandemic have refocused international attention on the importance of genomic surveillance and pandemic preparedness. Genomic surveillance of SARS-CoV-2 has played a critical role in understanding the impact of viral evolution on global transmission dynamics, pathogenesis, and therapeutic intervention. As discussed, although this pipeline has been continually refined during this pandemic, there is still much left to be improved on. Most notably, for an effective global response to a global pathogen, the gap between developed countries and LMICs needs to be addressed. Failure to properly allocate resources, build capacity, and foster local expertise in LMICs for genomic surveillance has been a long-standing issue that has continued through the COVID-19 pandemic. The lack of pre-existing networks for data sharing and the lack of standardized practices likewise are not unique to this pandemic and are larger issues that need to be addressed for future pandemic preparedness. The emergence of the virus from a yet unidentified animal reservoir further highlights the continued lack of genomic surveillance in animal species. As urbanization, globalization, and climate change increase the threat of zoonotic transmission and rapid spread, more resources should be dedicated to early detection of potential pandemic threats.

Regardless, no matter what improvements are implemented, genomic surveillance is inherently reactive; after the virus has mutated, researchers race to characterize its viral evolution and consequential phenotype. A proactive approach to understanding the role of viral variation on viral replication and immune escape could someday serve as a gold standard for an expedited understanding of phenotypic consequences. For example, Starr et al. experimentally measured how all amino acid mutations in the SARS-CoV-2 Spike protein would affect folded protein expression and its associated binding affinity to ACE2 to predict mutations that would enhance transmissibility [131]. Similarly, Obermeyer et al. developed a hierarchical Bayesian regression model to identify mutations associated with viral fitness measured by the growth rate of each lineage and arising mutations [132]. Likewise, Maher et al. used statistical modeling to identify the mutations that could result in future SARS-CoV-2 variants of concern [133]. Predicting and cataloging variants of concern before they arise would enable the preemptive design of therapeutics and public health measures to be employed at need. Much like efforts to prepare for future pandemic threats, proactive tracking would enable preparation against new pandemic variants.

## Figures and Tables

**Figure 1 viruses-14-02532-f001:**
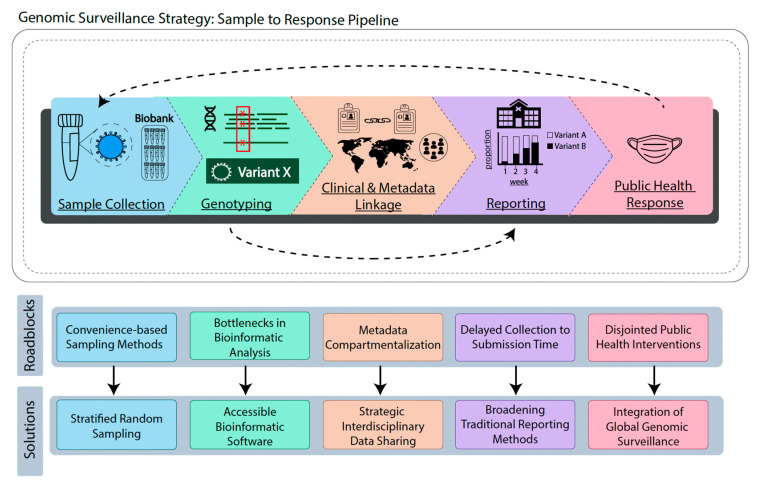
Molecular Surveillance Pipeline. The pipeline outlines the main concept of each process (sample collection, genotyping, clinical and metadata linkage, reporting, and public health response), and illustrates the solutions and roadblocks present at each stage of the pipeline. Asterisks in the genotyping section are representative of identified mutations in variant X.

**Figure 2 viruses-14-02532-f002:**
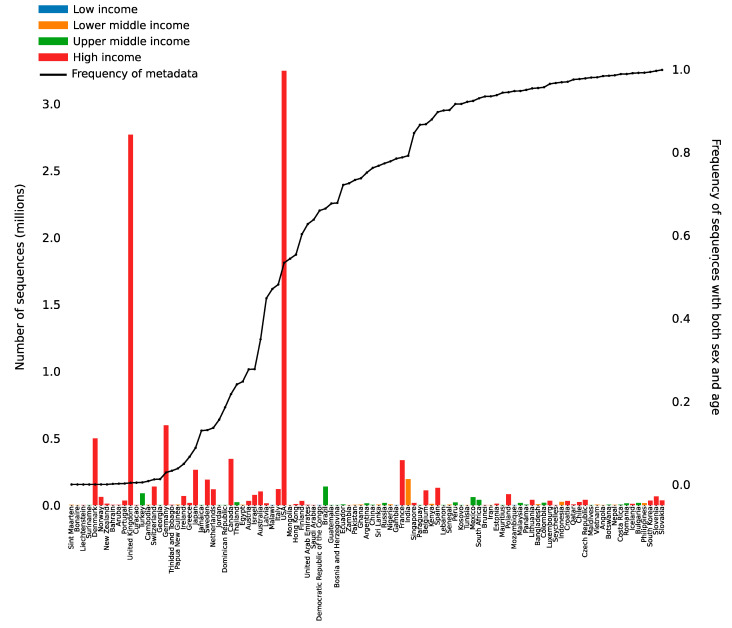
Sequence count and metadata prevalence by country. The number of SARS-CoV-2 whole-genome sequences available from each country in GISAID (accessed 5 September 2022) as colored bars. The colored bars represent the income level as defined by the Worldbank. The percentages of sequences with associated sex and age metadata are represented by the black line.

**Figure 3 viruses-14-02532-f003:**
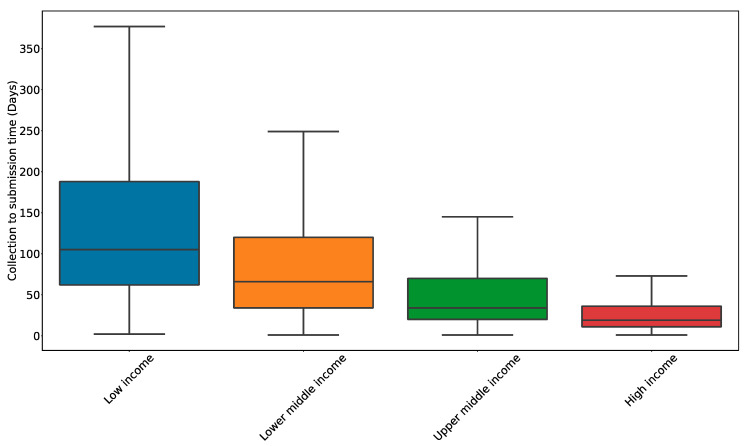
Lag time from specimen collection to sequence reporting on GISAID by income status. Data were pulled from GISAID (accessed 5 September 2022) and Worldbank and are presented as a boxplot. The box represents the lower and upper quartiles (Q1 and Q3, respectively) and the line within the box represents the median. The whiskers represent the rest of the data distribution except for the points deemed outliers (defined as 1.5 × IQR + Q3 or 1.5 × IQR − Q1 where IQR = Q3 − Q1).

**Figure 4 viruses-14-02532-f004:**
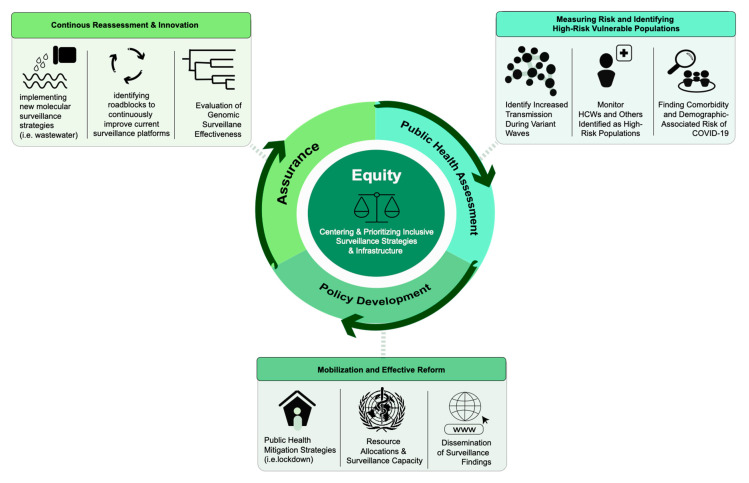
Public health framework and implications of genomic surveillance on public health responses. Framework based on the 10 essential public health services. This schematic represents each phase (assessment, policy development, assurance, and equity) in the context of SARS-CoV-2 genomic surveillance; HCW = healthcare worker.

**Figure 5 viruses-14-02532-f005:**
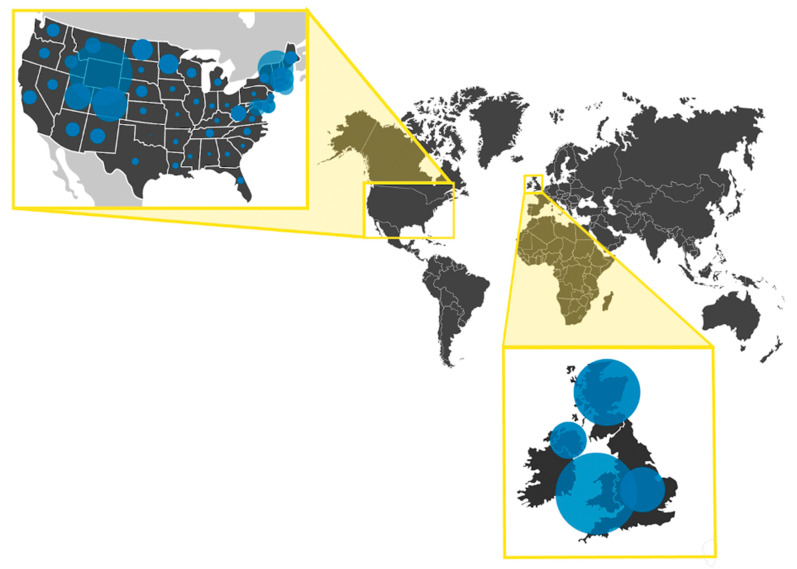
Relative sequence availability in the US and UK varies by region. The proportion of sequences from each state (USA) or region (UK) was calculated by dividing the sequence count from each region by the number of total sequences in the country (from GISAID as of 5 September 2022). Similar calculations were used to calculate the proportion of the population. The ratio of these two metrics is represented by the blue circles on the map.

## Data Availability

Publicly available datasets were analyzed in this study. Sequence data were downloaded from GISAID (https://gisaid.org/, accessed on 9 May 2022). Population data for the USA and UK were downloaded from the US Census Bureau (https://www.census.gov/, accessed on 9 May 2022) and the Office for National Statistics (www.ons.gov.uk, accessed on 9 May 2022), respectively. Economic status labels were downloaded from the Worldbank (https://data.worldbank.org/, accessed on 9 May 2022). Analyses were conducted in Python (v3.8.8) and figures were generated using Python package matplotlib (v3.3.4) or Adobe Illustrator.

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
