# Peer review of "Challenges and Opportunities for Global Genomic Surveillance Strategies in the COVID-19 Era"

_viruses, 2022, doi:10.3390/v14112532_

Round 1

Reviewer 1 Report

Hu et al present a study on the challenges and opportunities for global genomic surveillance strategies in the COVID-19 era. The authors discuss the different steps involved and challenges faced. Hu et al further provides potential solution by discussing case studies. The authors conclude with their assessment that proactive tracking would enable better prediction of pandemic threats.

As the title indicates, the study is extremely focused on the COVID-19 pandemic. This is also the main problem of this manuscript. Although COVID-19 was, and still is, a rather unique event in the recent history of pandemic threats, it was certainly not the first one in the era of genomic surveillance. The authors completely ignore any other efforts on tracking pathogen threats by genomic methods on a global scale. Influenza would be one extremely prominent example. The predecessor of the current "Bacterial and Viral Bioinformatics Resource Center", the "influenza Research Database" originated over 24 year ago at the Los Alamos National Laboratory. Although primarily for viral (and bacterial) sequence data, these resources were at the forefront decades before any COVID-19 surveillance efforts. Such program, not only for influenza but also other pathogens, such as HIV, M. tuberculosis or malaria are ongoing at various degrees. Similar assessments can be made with respect to clinical surveillance. The CDC in the United States as the WHO worldwide, for example, have multiple protocols in place for clinical surveillance in infectious diseases.

Overall, although the study by Hu et al. addresses interesting aspects of genomic surveillance strategies in the COVID-19 era, it tries to reinvent the wheel by disregarding earlier developments in decade long surveillance programs of other infectious diseases. Furthermore, the study then also lacks a "lessons learned" section or an analysis of repeated mistakes.

Author Response

The reviewer raises an excellent point. While our review is focused on SARS-CoV-2 genomic surveillance, we could have better contextualized the role of genomic surveillance in previous pathogen outbreaks. We have added a few lines of text to the Introduction (Lines 54 – 59) to better recognize this. In addition, our review highlights the success of previous surveillance systems such as the National Human Immunodeficiency Virus (HIV) Surveillance System and the National Notifiable Diseases Surveillance System (NNDSS) (Line 538 – 539). Furthermore, we identify timeliness, automation, and data completeness as critical components for successful pre-COVID-19 surveillance systems (Lines 356-362). Thus, while we intend our focus to be on SARS-CoV-2, we hope we now properly cite the influence of past efforts with other pandemic pathogens.

In addition, we have highlighted a few ‘cross-pandemic’ lessons in the Discussion (Lines 585-593) that highlight repeated mistakes in this pandemic. Thank you for your suggestions!

Reviewer 2 Report

This review paper described the limitation and future direction of genomic surveillance of SARS-CoV-2. The manuscript is well organized and discussed various aspects of the limitation of genomic surveillance. The study will contribute to the development of a global genomic surveillance system. I have only a few comments and some points need to be revised before publication.

1. The letters in figure 1 are illegible in the pdf file. The overall quality of Figure 1 needs to be revised.

2. Line 163-164: rRT-PCR and Spike gene Sarger sequencing can be used for detecting known variants, but it is not a real ‘Genotyping’. In addition, rRT-PCR based variant detection could not distinguish the recombinants

3. PANGOLIN program is the most popular program used in SARS-CoV-2 genotyping. It needs to be described in this review article.

4. While generating unprecedentedly large genomic data, many erroneous sequences have been uploaded to the database, becoming a huge obstacle in genomic analysis. There is a standard protocol for library preparation for NGS, but there is no standard for the assembly of NGS data. Nextclade programs have been used for quality control of sequencing results, but the program could not check raw NGS data. Authors need to discuss this limitation of sequencing quality.

5. Figure 2 needs to be more intuitive. If the number of sequences was represented by bars, the metadata prevalence needs to be represented by other forms.

6. section 6.4, 'equity' is a very important part, but it looks not related to section 6 'Public Health Response'. I think it needs to move to section 2 'Sample collection'

7. Due to the large outbreak, the ratio of the number of analyzed whole-genome / confirmed case numbers is very low. Is there any official recommendation for the ratio? or do the authors have any opinions?

Author Response

  1. The letters in figure 1 are illegible in the pdf file. The overall quality of Figure 1 needs to be revised.

We have increased both the letter size and the resolution of Figure 1 to increase the quality of the image.

  1. Line 163-164: rRT-PCR and Spike gene Sanger sequencing can be used for detecting known variants, but it is not a real ‘Genotyping’. In addition, rRT-PCR based variant detection could not distinguish the recombinants.

We agree with the reviewer that RT-qPCR methods for distinguishing between lineages have distinct limitations, but are important tools in resource-limited settings. To be sure that we discuss these drawbacks fully, we have added some additional qualifications to the use of these tools for genotyping purposes (Lines 235-250).

  1. PANGOLIN program is the most popular program used in SARS-CoV-2 genotyping. It needs to be described in this review article.

We thank the reviewer for pointing out this oversight; PANGOLIN is an important tool to discuss. We have added a discussion of this nomenclature and its complementarity to other discussed systems (Lines 167-173).

  1. While generating unprecedentedly large genomic data, many erroneous sequences have been uploaded to the database, becoming a huge obstacle in genomic analysis. There is a standard protocol for library preparation for NGS, but there is no standard for the assembly of NGS data. Nextclade programs have been used for quality control of sequencing results, but the program could not check raw NGS data. Authors need to discuss this limitation of sequencing quality.

The reviewer raises a good point and we recognize these challenges from some of our own work. A number of studies have been published that explore this issue and that suggest a few different solutions, including conducting further benchmarking and assessment of genome assemblers, and by the use of multiple, independent assemblers when doing analysis (https://www.liebertpub.com/doi/full/10.1089/omi.2022.0042). Other solutions have been proposed, including the implementation of novel metrics (https://journals.asm.org/doi/10.1128/JCM.00944-21) that can alleviate some of the standardization of raw sequence quality. Finally, cutoff thresholds (as proposed by https://www.science.org/doi/10.1126/scitranslmed.abe2555 or https://pubmed.ncbi.nlm.nih.gov/33813118/) could also be used as means of QC before the sequence is input into Nextclade. While we fear some of these suggestions may be too specialized for the audience of this review, we have added a broader discussion of this issue and have referenced the above papers (Lines 217-219 and 221-223).

  1. Figure 2 needs to be more intuitive. If the number of sequences was represented by bars, the metadata prevalence needs to be represented by other forms.

We thank the reviewer for this note and have modified the figure with the frequency of metadata represented as a line and the bars representing the number of sequences per country, colored by income status as defined by the World Bank. We hope this is more intuitive!

  1. section 6.4, 'equity' is a very important part, but it looks not related to section 6 'Public Health Response'. I think it needs to move to section 2 'Sample collection'.

Although we agree that equity is a crucial consideration in sample collection, our framework for the ‘Public Health Response’ section highlights equity as central to assessment, policy development, and assurance. This decision is supported by the ‘10 essential public health services’ framework referenced in the text and figure. Thus, while we opted to keep the section in place, we did note equity’s importance in section 2 and emphasized that it will be revisited in section 6.4. (Line 143 – 145).

  1. Due to the large outbreak, the ratio of the number of analyzed whole-genome / confirmed case numbers is very low. Is there any official recommendation for the ratio? or do the authors have any opinions?

We thank the reviewer for this thoughtful question and, though we address sampling bias, we do not provide a recommendation for ratio of sequences to cases. Many modeling approaches have tried to address this question, such as https://www.medrxiv.org/content/10.1101/2021.01.12.21249613v1, which suggests 5% of all positive tests could detect variants when they are circulating at 0.1% to 1%, or https://www.medrxiv.org/content/10.1101/2021.12.30.21268453v2.full#ref-8, which tries to provide sample size calculations while addressing different biases in their modeling approach to account for biological and technical biases. However, most countries do not have the infrastructure and sequencing capabilities to achieve these numbers, so we believe that an emphasis should rather be put on representative sampling strategies (addressed in Lines 73-162).

Reviewer 3 Report

Dear Authors, 

I greatly enjoyed reviewing your article entitled "Challenges and opportunities for global genomic surveillance strategies in the COVID-19 Era". It is an excellent review article, with clear figures, and an extensive bibliography, written in excellent an easy-to-read English. The article is spot on, and I recognize many challenges encountered in participating to the setting up SARS-CoV-2 genomic surveillance in our country, such as lack of incentives to share specimens for surveillance, compartimentalization and lack of interoperability. 

I have to minor comments: 

1) Page 2, line 51: I am not sure that the word "metadatal" can be used. Please reconsider;

2) Page 3, line 99: there seems to have a glitch with your reference management system there (what do "33808716, 32645347" mean ? I guess these are two references not correctly ordered);

Congratulations for your article and thanks for the opportunity to review. 

Best regards, 

Author Response

1) Page 2, line 51: I am not sure that the word "metadatal" can be used. Please reconsider.

We thank the reviewer for spotting this error and have corrected it to “data” (Line 61).

2) Page 3, line 99: there seems to have a glitch with your reference management system there (what do "33808716, 32645347" mean ? I guess these are two references not correctly ordered).

We again thank the reviewer for their careful review and have inserted the proper references (Line 104).

Congratulations for your article and thanks for the opportunity to review. 

Thank you!

Round 2

Reviewer 1 Report

The authors addressed all the raised concerns.